# Fat Quantity and Quality, as Part of a Low-Fat, Vegan Diet, Are Associated with Changes in Body Composition, Insulin Resistance, and Insulin Secretion. A 16-Week Randomized Controlled Trial

**DOI:** 10.3390/nu11030615

**Published:** 2019-03-13

**Authors:** Hana Kahleova, Adela Hlozkova, Rebecca Fleeman, Katie Fletcher, Richard Holubkov, Neal D. Barnard

**Affiliations:** 1Physicians Committee for Responsible Medicine, 5100 Wisconsin Ave, N.W. Ste.400, Washington, DC 20016, USA; adela.hlozkova@gmail.com (A.H.); rfleeman95@gmail.com (R.F.); katie.fletcher@yale.edu (K.F.); nbarnard@pcrm.org (N.D.B.); 2School of Medicine, University of Utah, Salt Lake City, UT 84132, USA; richard.holubkov@hsc.utah.edu; 3Adjunct Faculty, George Washington University School of Medicine and Health Sciences, Washington, DC 20016, USA

**Keywords:** diet, nutrition, plant-based, fat, fatty acids, vegan

## Abstract

Macronutrient composition of the diet influences the development of obesity and insulin resistance. The aim of this study was to assess the role of dietary fat quantity and fatty acid composition in body composition, insulin resistance, and insulin secretion. An open parallel randomized trial design was used. Overweight participants (*n* = 75) were randomized to follow a low-fat vegan (*n* = 38) or control diet (*n* = 37) for 16 weeks. Dual X-ray absorptiometry was used to measure body composition. Insulin resistance was assessed with the Homeostasis Model Assessment (HOMA-IR) index. Insulin secretion was assessed after stimulation with a liquid breakfast (Boost Plus, Nestle, Vevey, Switzerland). Self-reported 3-day diet records were used to assess dietary intake. A linear regression model was used to test the relationship between fat intake and body composition, insulin resistance, and insulin secretion. Changes in fat intake expressed as percent of total energy consumed correlated positively with changes in fat mass (*r* = 0.52; *p* < 0.001; and 0.347; *p* = 0.006, respectively), even after adjustment for changes in body-mass index (BMI) and energy intake (0.33; *p* = 0.01). Decreased intakes of C18:0 (*r* = 0.37, *p* = 0.004) and CLA-trans-10-cis12 (*r* = 0.40, *p* = 0.002), but increased intake of C18:2 (*r* = −0.40, *p* = 0.002) and C18:3 (*p* = −0.36, *p* = 0.006), were associated with a decrease in HOMA-IR, independent on changes in BMI and energy intake. The main fatty acids associated with changes in fasting insulin secretion were C12:0 (*r* = −0.31, *p* = 0.03), and TRANS 16:1 (*r* = −0.33, *p* = 0.02), both independent on changes in BMI and energy intake. Our findings demonstrate that, in the context of a low-fat vegan diet, decreased intake of saturated and trans fats and increased relative content of polyunsaturated fatty acids, particularly linoleic and α-linolenic acids, are associated with decreased fat mass and insulin resistance, and enhanced insulin secretion.

## 1. Introduction

Dietary macronutrients play an important role in the risk of obesity and type 2 diabetes. In particular, the quality of fat is reflected in the composition of plasma and tissue lipids and, in turn, affects cell membrane function, including insulin signaling [1,2].

Observational studies have indicated that dietary fat quality may be related to the development of insulin resistance and metabolic syndrome, independent of effects on body weight [3]. Intervention studies have shown that substituting unsaturated (polyunsaturated and/or monounsaturated) fat for saturated fat in the diet alters plasma fatty acids, resulting in increased insulin sensitivity [4]. Among polyunsaturated fats, increased dietary intake of linoleic acid (C18:2) improves insulin sensitivity. On the other hand, long-chain n-3 fatty acid supplements do not appear to improve insulin sensitivity or glucose metabolism [5].

Low plasma concentrations of linoleic acid and high plasma concentrations of palmitic acid are potential predictors of the risk of developing diabetes mellitus [6]. A high proportion of linoleic acid in plasma fatty acids, indicating a high intake of dietary linoleic acid, has been associated with a lower risk of developing diabetes [7]. As the Western diet has a low proportion of linoleic acid, which is reflected in low proportions of linoleic acid in plasma phospholipids, some have suggested favoring vegetable fat over animal fat as a means of preventing diabetes [8].

Several observational studies have assessed fatty acid intake in people consuming plant-based diets. A Danish study showed that the intake of fatty acids and the ratio of polyunsaturated to saturated fatty acids were more favorable among those adhering to a vegan diet than an omnivorous diet [9]. Specifically, participants on vegan diets had a lower mean intake of saturated fatty acids and a much higher intake of linoleic acid, compared with omnivores [10]. Similarly, the European Prospective Investigation into Cancer and Nutrition-Oxford study showed that individuals on vegan diets had very low intake of saturated fat and high intake of polyunsaturated fatty acids, compared with omnivores, pesco-vegetarians, and lacto-ovo-vegetarians [11].

It has been shown previously that a plant-based diet increases the proportion of linoleic acid (C18:2) in serum phospholipids, and that this increase was related to increased insulin sensitivity in patients with type 2 diabetes [12].

The overall aim of this 16-week randomized clinical trial was to assess the effect of diet changes on body weight and metabolism. Here, we explored the role of specific dietary fats on body composition, insulin resistance, and insulin secretion in overweight individuals with no history of diabetes. Our hypothesis was that reduced consumption of saturated fat and increased relative intake of linoleic acid (C18:2) induced by a vegan diet would be related to reduced body fat mass and increased insulin sensitivity.

## 2. Materials and Methods

### 2.1. Study Design

The study was a single-center, randomized, open parallel study conducted between October 2016 and June 2017. Adult men and women, with a body-mass index (BMI) between 28 and 40 kg/m^2^, were enrolled. Exclusion criteria were history of diabetes, smoking, alcohol or drug abuse, pregnancy or lactation, and current use of a vegan diet. The study protocol was approved by the Chesapeake Institutional Review Board on October 12, 2016. The protocol identification number is Pro00018983. All participants gave written informed consent. Trial registration: ClinicalTrials.gov number: NCT02939638

### 2.2. Randomization and Study Groups

Using a 1:1 ratio, participants were randomly assigned to the vegan or the control group, and examined at baseline and 16 weeks. Participants in the vegan group were asked to adhere to a low-fat vegan diet consisting of vegetables, grains, legumes, and fruits. They were also asked to avoid added oils. The daily allowance for fat intake was set between 20 and 30 g. This level of intake ensured that participants were able to consume adequate amounts of the essential fatty acids. No meals were provided. The control group participants were asked to maintain their current diets for the full 16 weeks.

### 2.3. Dietary Intake, Physical Activity, and Medications

After teaching the participants how to give detailed reports, each participant completed a 3-day dietary record at baseline and 16 weeks. Dietary intake data were collected and analyzed by a registered dietician, using Nutrition Data System for Research version 2016, developed by the Nutrition Coordinating Center (NCC), University of Minnesota, Minneapolis, MN [13]. Dietary adherence was also checked over the phone by doing random periodic phone calls to evaluate an over-the-phone food record with the participants. Physical activity was assessed by the International Physical Activity Questionnaire (IPAQ) [14]. All study participants were instructed to maintain their pre-existing medication regimens for the duration of the study (unless otherwise instructed by their personal physicians).

### 2.4. Outcomes

The main outcomes were fat mass, insulin secretion, and insulin resistance. Following a 12-h overnight water-only fast, all measurements were performed on an outpatient basis at baseline and 16 weeks. Fat mass was assessed using a Dual X-ray absorptiometry(DXA) scan (iDXA; GE Healthcare, Chicago, IL, USA). Insulin secretion was assessed after stimulation with a liquid breakfast (Boost Plus, Nestle, Vevey, Switzerland; 720 kcal, 34% of energy from fat, 16% protein, 50% carbohydrate). Plasma concentrations of glucose, immunoreactive insulin, and C-peptide were measured at 0, 30, 60, 120, and 180 min. Serum glucose was analyzed using the Hexokinase UV endopoint method (Roche, Basel, Switzerland). Plasma immunoreactive insulin and C-peptide concentrations were determined using insulin and C-peptide electro-chemiluminescence immunoassay (ECLIA) kits (Roche, Basel, Switzerland), respectively. HbA1c was measured by turbidimetric inhibition immunoassay (Roche, Basel, Switzerland). Plasma lipids concentrations were measured by enzymatic colorimetric methods (Roche, Basel, Switzerland). Insulin resistance was calculated as HOMA-IR (Homeostasis Model Assessment) index [15].

### 2.5. Exposure Variables

Dietary intake of fat and the intake of specific fatty acids, as calculated from dietary records, were used as predictors of changes in body composition, insulin secretion, and insulin resistance.

### 2.6. Statistical Analysis

We based a calculation of the sample size on an alpha of 0.05 and 0.80 beta to detect between-group differences in outcome variables: a clinically relevant 10% difference in insulin resistance (HOMA-IR) and a 5% difference in body weight. It indicated the need for 54 participants to complete the intervention to which they were assigned. The intention-to-treat analysis included all participants. A repeated measure ANOVA model was used to test the between-group differences from baseline to 16 weeks. Factors group, subject, and time were included in the model. Interaction between group and time (Gxt) was calculated for each variable. Within each diet group, paired comparison t-tests were calculated to test whether the change from baseline to 16 weeks was significantly different from zero. The level of significance was 0.05. Regression analyses assessed the effect size of changes in fat intake and of specific fatty acid intake on body composition, insulin secretion, and insulin resistance.

## 3. Results

### 3.1. Characteristics of the Participants

The flow of participants through the study is shown in Figure 1. Out of 75 participants who were randomized, 96% (*n* = 72) completed the entire study. The mean age was 53.2 ± 12.6 years. The vast majority of our study participants (89%, *n* = 67) were women. The baseline characteristics of the study population are shown in Table 1.

### 3.2. Physical Activity and Dietary Intake

Physical activity and dietary intake are shown in Table 2. Physical activity did not change substantially in either group. Both groups reduced reported energy intake, with no significant difference between groups (*p*-value for interaction between the factors group and time, Gxt, *p* = 0.69). Mean intake of carbohydrate, total fat, and protein did not change significantly in control participants, but there was a significant reduction in their mean saturated fatty acid intake (*p* = 0.002). The vegan group participants increased their mean intake of carbohydrate (*p* < 0.001) and fiber (*p* < 0.001), while decreasing consumption of total protein (*p* < 0.001), fat (*p* < 0.001), and cholesterol (*p* < 0.001).

### 3.3. Changes in Fatty Acid Intake

Absolute changes in fatty acid intake expressed in g/day are listed in Table 2. After the 16-week dietary intervention, the vegan group had reduced its fat intake from 36.1% of total energy to 17.5% of total energy, with total grams of fat per day decreasing on average by 48.9 g (from 77.7 to 28.8 g). The most significant changes in fatty acid intake were observed in palmitic acid, stearic acid, oleic acid, and linoleic acid (−5.69, −2.54, −12.75, and −3.47 g in average, respectively). For the control group, mean fat intake was 35.5% of energy at baseline and 35.0% at 16 weeks. The relative changes in fatty acid intake expressed as percent of total fat intake are shown in Table 3 and Figure 2. The fatty acid composition of the diet did not change significantly in the control group, while a significant reduction in saturated fatty acids and a relative increase in polyunsaturated fatty acid intake were observed in the vegan group.

### 3.4. Body Weight, Body Composition, Insulin Resistance, and Insulin Secretion

Changes in anthropometric variables are shown in Table 2. Body weight decreased significantly only in the vegan group (treatment effect −6.5; 95% CI −8.9 to −4.1 kg; Gxt, *p* < 0.001). Fat mass and, particularly, visceral fat volume were reduced only in the vegan group (treatment effect −4.3; 95% CI −5.4 to −3.2 kg; Gxt, *p* < 0.001; and treatment effect −224; 95% CI −328 to −120 cm^3^; Gxt, *p* < 0.001, respectively). HOMA-IR was reduced significantly only in the vegan group (treatment effect −1.0; 95% CI −1.2 to −0.8; Gxt, *p* = 0.004). A dose-response increase in insulin secretion as a function of plasma glucose concentrations was observed in the intervention group compared with controls (Gxt, *p* < 0.001), and was reported in detail elsewhere [16].

### 3.5. The Relationship between Fat Intake and Body Composition

The changes in percent dietary energy in the form of fat were positively associated with changes in BMI (*r* = 0.51; *p* < 0.001). As percent energy from fat decreased, fat mass and visceral fat volume also decreased (*r* = 0.52; *p* < 0.001; Figure 3A; and 0.347; *p* = 0.006, respectively). The association between changes in percent energy in the form of fat and percent of body fat remained significant even after adjustment for changes in BMI and changes in energy intake (*r* = 0.33; *p* = 0.011).

We observed a positive association between changes in C14:0, C16:0, and C18:0, and changes in fat mass (*r* = 0.30, *p* = 0.01; *r* = 0.37, *p* = 0.002; and 0.34, *p* = 0.005, respectively). As the intake of these saturated fatty acids decreased, the fat mass also decreased. These associations were no longer significant after adjustments for changes in BMI and changes in energy intake. Finally, a positive association was observed between changes in TRANS 18:2 and changes in the percentage of body fat. The lower the TRANS 18:2 intake, the lower the percentage of body fat (*r* = 0.35; *p* = 0.004; Figure 3B). This association remained significant even after adjustment for changes in BMI and energy intake (*r* = 0.35; *p* = 0.007).

### 3.6. The Relationship between Fatty Acids and Insulin Resistance

The changes in percent energy in the form of fat were positively associated with changes in HOMA-IR (*r* = 0.30; *p* = 0.02). However, after adjustment for changes in BMI and for changes in energy intake, this association was no longer significant. As the percent intake of C18:0 and CLA-trans10-cis12 decreased, HOMA-IR decreased (*r* = 0.37, *p* = 0.004; Figure 3C; and *r* = 0.40, *p* = 0.002; Figure 3D, respectively). These associations remained significant even after adjustment for changes in BMI and for changes in energy intake (*r* = 0.27, *p* = 0.04; and *r* = 0.32, *p* = 0.02, respectively). A negative relationship was observed between changes in C18:2 and C18:3 and HOMA-IR. As the percent intake of C18:2 and C18:3 increased, HOMA-IR decreased (*r* = −0.40, *p* = 0.002; Figure 3E, and *p* = −0.36, *p* = 0.006; Figure 3F, respectively). These associations remained significant even after adjustment for changes in BMI and for changes in energy intake (*r* = −0.30, *p* = 0.03; and r = −0.27, *p* = 0.049, respectively).

### 3.7. The Relationship between Fatty Acids and Insulin Secretion

A negative association between changes in the intake of C8:0, C12:0, C14:0, and TRANS 16:1 with changes in fasting insulin secretion was observed (*r* = −0.29, *p* = 0.03; *r* = −0.31, *p* = 0.03; Figure 3G; *r* = −0.28, *p* = 0.04, and *r* = −0.33, *p* = 0.02; Figure 3H, respectively). As the intake of these saturated fatty acids decreased, the fasting insulin secretion increased. The association remained significant after adjustments for changes in BMI and changes in energy intake for C12:0 and TRANS 16:1 (*r* = −0.29, *p* = 0.04 and *r* = −0.30, *p* = 0.03, respectively).

There was a negative association between changes in dietary C20:5, C22:5, and C22:6, and changes in postprandial insulin secretion. As the intake of these fatty acids decreased, the postprandial (60 min) insulin secretion increased (*r* = −0.28, *p* = 0.03; *r* = −0.29, *p* = 0.02; and *r* = −0.33, *p* = 0.009, respectively). These associations remained significant even after adjustments for changes in BMI and changes in energy intake (*r* = −0.27, *p* = 0.04; *r* = 0.29, *p* = 0.03; and *r* = −0.32, *p* = 0.01, respectively).

## 4. Discussion

### 4.1. Main Findings

This 16-week randomized controlled study demonstrated that, in overweight individuals, both fat quantity and fat quality were associated with changes in body weight, body composition, insulin resistance, and insulin secretion. Decreased intake of saturated, trans, or total fat was associated with decreased fat mass. Changes in fatty acid composition of the diet were associated with changes in insulin resistance and insulin secretion. Decreased intakes of C18:0 and CLA-trans-10-cis12, but increased intake of C18:2 and C18:3, were associated with a decrease in insulin resistance, independent on changes in BMI and independent on changes in energy intake.

### 4.2. Metabolic Effects of Saturated Fatty Acids

Our study has demonstrated a positive relationship between intake of stearic acid (C18:0) and insulin resistance, and a negative association between intake of lauric acid (C12:0) and fasting insulin secretion. These findings from this randomized trial are consistent with those of observational studies. A 2016 analysis of data from two large prospective U.S. cohorts, that included 73,147 women from the Nurses’ Health Study and 42,635 men from the Health Professionals Follow-up Study, showed that dietary intakes of major individual saturated fatty acids, including lauric acid (C12:0), myristic acid (C14:0), palmitic acid (C16:0), and stearic acid (C18:0), were positively associated with risk of coronary heart disease. Replacement of 1% daily energy intake from these four saturated fatty acids combined by equivalent energy from polyunsaturated fat, whole grains, or plant protein, was associated with a 6–8% reduced risk of coronary heart disease [17].

Some studies have also established a link between C18:0 and an increased risk of cardiometabolic disease [18,19]. A meta-analysis of 60 trials assessed the effects of dietary fatty acids on plasma lipids. Lauric acid (C12:0) was found to greatly increase total plasma cholesterol and to impair the blood lipid profile, while myristic (C14:0), palmitic (C16:0), and stearic (C18:0) acids had a less pronounced negative effect on blood lipids. The authors of the meta-analysis concluded that cardiovascular risk may be reduced most effectively when trans and saturated fatty acids are replaced with cis unsaturated fatty acids [18]. Furthermore, saturated fat intake was inversely related to diabetes incidence in the PREDIMED study, a prospective study that analyzed data from 3349 people who were followed for 4.3 years. Participants in both the Mediterranean diet groups and control group in the highest quartile of animal fat had double the risk of type 2 diabetes, compared with their counterparts in the lowest quartile [20].

### 4.3. Metabolic Effects of Trans Fatty Acids

Even though the use of trans fatty acids is declining, they are still present in some processed foods, ruminant fat (meat and dairy products), and partially hydrogenated vegetable oils [21,22,23]. Our findings show a positive association between dietary intake of TRANS 18:2 and percentage body fat, and a positive relationship between consumption of CLA-trans10-cis12 and insulin resistance. These results confirm the findings of observational studies. Data from the Nurses’ Health Study, a large prospective study that analyzed data from 41,518 women who were followed for 8 years, showed that increased trans fatty acid intake was associated with weight gain. This association was even more pronounced in overweight women [24].

A 2011 randomized trial examined the effect of a high intake of trans fatty acids on fat deposition and blood lipids in overweight postmenopausal women. The women were randomized to receive either partially hydrogenated soybean oil providing 15.7 g/day of trans fatty acids or a control oil with mainly oleic and palmitic acid. After a 16-week intervention, a high trans-fat intake predictably increased LDL-cholesterol by 18% and decreased HDL-cholesterol by 10%, but also tended to increase the body fat and waist circumference compared with the control fat [25].

In observational studies, consumption of trans fatty acids has been associated with coronary heart disease [21,26] and diabetes mellitus [22]. A 2018 study examined the association between plasma concentrations of the four major trans fatty isomers and diabetes using data from the National Health and Nutrition Examination Survey (NHANES). Higher plasma concentrations of total trans fatty acids, in particular elaidic acid (C18:1 n-9t), palmitelaidic acid (C16:1 n-7t), and linolelaidic acid (C18:2 n-6t, 9t), were associated with diabetes and biomarkers of altered glucose metabolism in U.S. adults [22]. Our data show that the dietary intake of trans fatty acids, especially trans 18:2 (linolelaidic acid), was positively associated with changes in percentage of body fat. Our study provides a potential explanation for a higher risk of diabetes and myocardial infarction associated with the consumption of this trans fatty acid [21,22].

Our study also showed a positive association between dietary intake of CLA-trans10-cis12 and insulin resistance. This finding is in accord with multiple studies that have investigated the effect of CLA-trans10-cis12, a trans fatty acid naturally found in dairy and beef fat, on markers of glucose and lipoprotein metabolism in humans [27,28,29]. A 2002 randomized, double-blind, controlled trial investigated the effect of CLA-trans10-cis12 on insulin sensitivity, lipid metabolism, and body composition in 60 abdominally obese men with signs of the metabolic syndrome, who were treated with 3.4 g/day CLA (isomer mixture), purified CLA-trans10-cis12, or a placebo. Higher intake of CLA-trans10-cis12 was associated with insulin resistance, hyperglycemia, and dyslipidemia [27]. Similarly, a 2004 randomized, double-blind, placebo-controlled trial found that CLA-trans10-cis12 supplementation significantly increased fasting glucose concentrations and reduced insulin sensitivity [28].

CLA-trans10-cis12 decreases fatty acid uptake and utilization of exogenously derived fatty acids, thereby limiting substrate utilization in triglycerides synthesis. Chronic treatment with CLA decreases PPARγ expression, thus inhibiting glucose and fatty acid uptake and metabolism [29]. CLA-trans10-cis12 inhibits the differentiation of human preadipocytes into mature insulin-sensitive adipocytes. It promotes whole-body insulin resistance, leading to hyperglycemia and hyperlipidemia [27].

### 4.4. Metabolic Effects of Polyunsaturated Fatty Acid (PUFA)

Our study demonstrated an inverse relationship between relative intakes of linoleic (C18:2) and α-linolenic (C18:3 n3) acids and insulin resistance. The relative consumption of both of these fatty acids as a proportion of total fat significantly increased on the vegan diet, and insulin resistance fell proportionately. These findings agree with those of observational studies that have identified low levels of linoleic acid in serum phospholipids as a potential biomarker for diabetes risk [6,7]. A 2018 pooled analysis from 20 prospective studies from 10 countries confirmed long-term benefits of linoleic acid for the prevention of diabetes [30]. Therefore, consumption of fat contained in plant foods has been suggested as one of the means to influence diabetes risk [8].

It has also been shown that dietary intake of α-linolenic acid is associated with beneficial effects on cardiometabolic risk factors and insulin sensitivity, thereby reducing the progression to cardiovascular disease and type 2 diabetes [31]. A 2016 cross-sectional study assessed the association between adipose tissue fatty acids (α-linolenic acid, EPA, DHA) and insulin resistance in participants from the Adventist Health Study-2 cohort. Higher adipose tissue α-linolenic acid content was inversely associated with insulin resistance in healthy adults, while no associations were noted for adipose tissue EPA plus DHA and insulin resistance [32]. These results are consistent with another cross-sectional study from Japan, which found a significant association of higher intake of α-linolenic acid with a lower prevalence of insulin resistance in normal weight individuals [33].

Our results also show an inverse association between intake of EPA, DHA and DPA, and postprandial insulin secretion. The lack of benefits of omega-3 fatty acids on incidence of type 2 diabetes has been summarized in the 2012 systematic review and meta-analysis of 16 studies comprising 540,184 participants [34]. It has even been shown that EPA and DHA supplementation may increase HbA1c in patients with type 2 diabetes [35]. Current evidence does not support increased intake of EPA and DHA in the prevention or treatment of type 2 diabetes, and our results provide additional confirmation and metabolic insights.

### 4.5. Metabolic Effects of Monounsaturated Fatty Acid (MUFA)

Our results did not show any positive metabolic effects of MUFA intake. Although a 2016 systematic review and meta-analysis of randomized controlled trials demonstrated metabolic benefits of MUFA-enriched diets, namely a decrease in body weight, in fasting plasma glucose, triglycerides, and systolic blood pressure, and increase in HDL-cholesterol [36], data from the Nurses’ Health Study suggest that the cardiovascular benefits associated with MUFA consumption may be expressed only when derived from plant-based foods, not from animal sources [37]. Therefore, consuming MUFA may bring cardiometabolic benefits due to substituting saturated fat from animal foods rather than having direct insulin-sensitizing effects.

### 4.6. Strengths and Limitations

There are several strengths to this study. It used a randomized parallel design. The simultaneous start date of the dietary intervention ensured that results were not affected by seasonal fluctuations in the diet. The study also provides the advantage of applicability beyond the research setting. The duration of the study was sufficient for participants to adjust to the new diet and adapt their cooking and restaurant habits accordingly. The attrition rate (4%) was low, which not only provides confidence in the interpretation of results, but also suggests that the dietary intervention may be acceptable and sustainable, as suggested by previous studies [38]. It also has important limitations. In free-living individuals, the reporting of dietary intake can be inaccurate. The self-reported 3-day dietary records completed at baseline and at 16 weeks may have been compromised by under-reporting or mistakes in recall [39]. Also, these records may not have been fully representative of the diets consumed throughout the intervention. However, we attempted to attenuate any discrepancies by teaching participants how to give detailed reports and by doing random periodic phone calls to evaluate an over-the-phone food record with participants. We did not analyze the fatty acid composition of plasma phospholipids to verify the dietary changes. Nonetheless, the observed weight loss and positive metabolic changes reassure us that the reported changes in nutrient intake were generally accurate. Lastly, the majority of the individuals who participated did indeed adhere to the diet intervention guidelines. It is likely that this research population represents individuals who are specifically interested in weight loss, rather than the population in general.

### 4.7. Potential Mechanisms

The vegan diet resulted in a significant decrease in fat and protein intake and an increase in carbohydrate consumption. The changes in macronutrient intake were associated with changes in body weight, body composition, and insulin resistance. Furthermore, changes in micronutrient intake may have influenced lipoprotein composition, transport, and metabolism. It has been shown previously that a plant-based diet significantly lowers body fat and circulating leptin levels in healthy volunteers [40] and in people with type 2 diabetes [41]. Leptin is one of the regulating hormones involved in appetite and weight regulation and its changes may be partly responsible for the beneficial changes of a plant-based diet on body weight and body composition in our study. It has been demonstrated previously that a plant-based diet reduces oxidative stress, which is another potential mechanism of influencing insulin resistance [41].

### 4.8. Practical Implications

Not only was a decrease in fat consumption associated with a decrease in percent of body fat, but limiting the intake of saturated and trans fats and increasing the relative content of polyunsaturated fatty acids, particularly linoleic and α-linolenic acids, was associated with increased insulin sensitivity and enhanced insulin secretion. Different types of dietary fats not only have different short-term metabolic effects, but also different associations with total mortality. These clinical trial findings resonate with those of large observational studies. A 2016 analysis from 83,349 women from the Nurses’ Health Study and 42,884 men from the Health Professional Follow-up Study showed that replacing 5% of energy from saturated fats with equivalent energy from polyunsaturated and monounsaturated fats was associated with estimated reductions in total mortality of 27% and 13%, respectively [42].

## 5. Conclusions

In the frame of a 16-week randomized controlled study, decreased fat intake, particularly saturated and trans fat, was associated with decreased fat mass. Decreased intakes of C18:0 and CLA-trans-10-cis12, and increased intake of C18:2 and C18:3, were associated with a decrease in insulin resistance, independent on changes in BMI and independent on changes in energy intake. The main fatty acids associated with unfavorable changes in fasting insulin secretion were C12:0 and TRANS 16:1, both independent on changes in BMI and independent on changes in energy intake. Selecting foods so as to limit the intake of saturated and trans fats and increase the relative content of polyunsaturated fatty acids, particularly linoleic and α-linolenic acids, may be a useful strategy for metabolic health.

## Figures and Tables

**Figure 1 nutrients-11-00615-f001:**
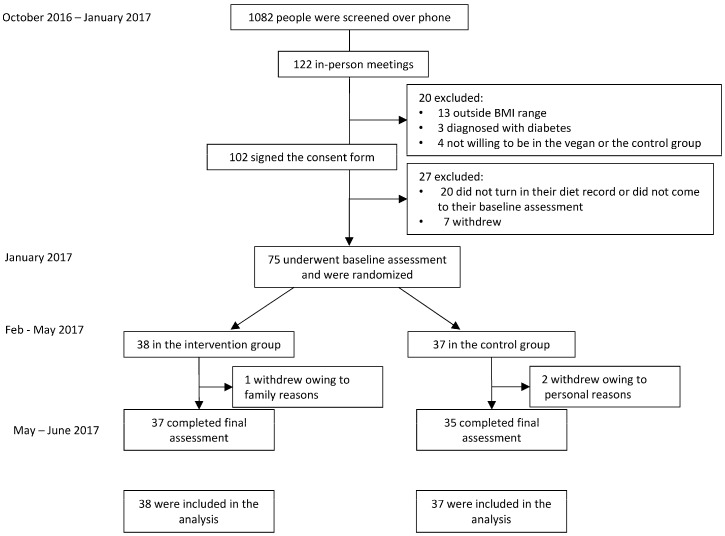
Enrollment of the Participants and Completion of the Study.

**Figure 2 nutrients-11-00615-f002:**
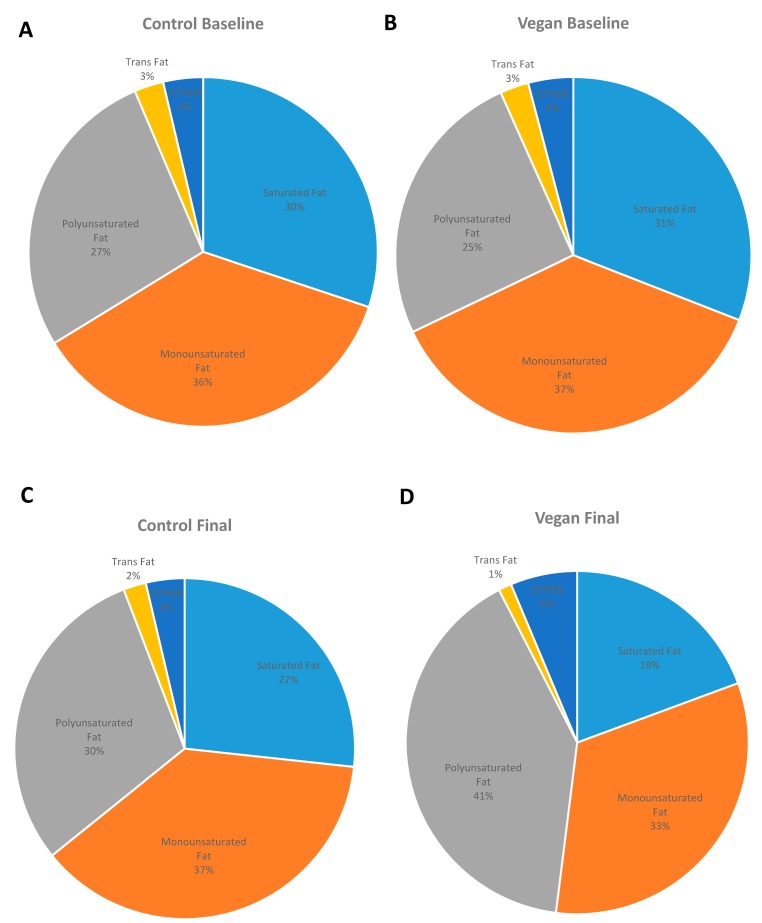
Changes in fatty acid profiles between vegans and controls at baseline and 16 weeks. (**A**) Control Baseline, (**B**) Vegan Baseline (**C**) Control Final, (**D**) Vegan Final. MUFA, monounsaturated fatty acid; PUFA, polyunsaturated fatty acid.

**Figure 3 nutrients-11-00615-f003:**
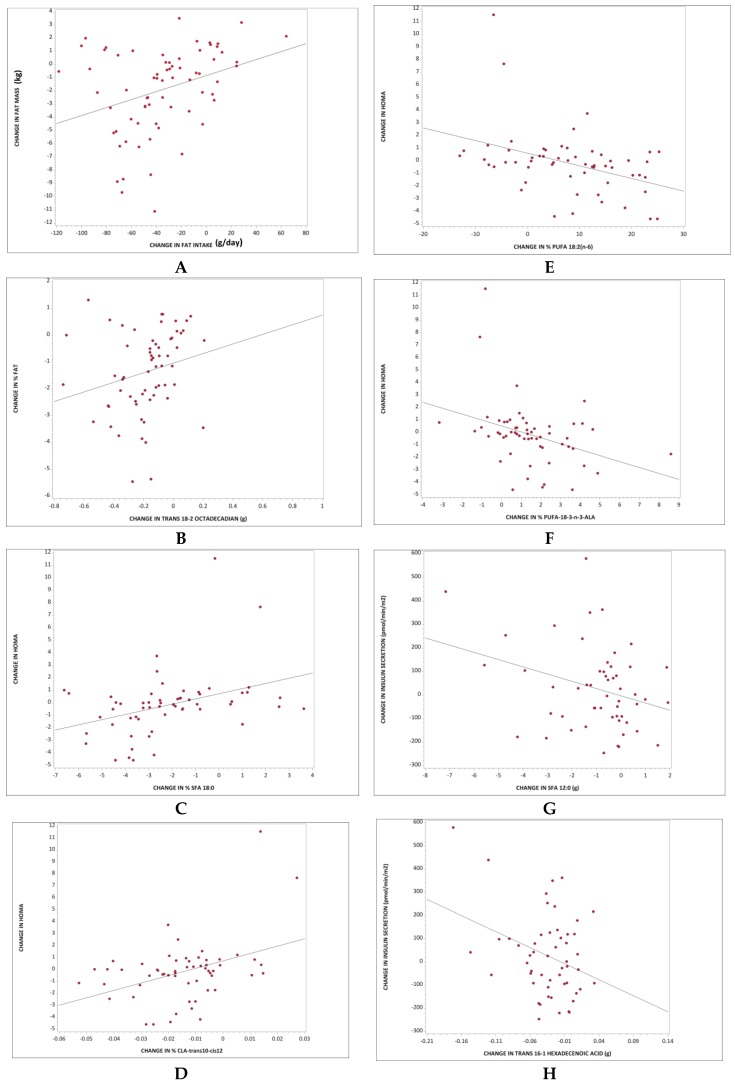
Regression models for changes in fatty acid intake and changes in body composition, insulin resistance, and insulin secretion. (**A**) Total fat intake and change in fat mass: *r* = 0.52; *p* < 0.001; (**B**) Intake of TRANS 18:2 and change in percentage of body fat: *r* = 0.35; *p* = 0.004; (**C**) Intake of C18:0 and changes in Homeostasis Model Assessment (HOMA-IR) index: *r* = 0.37, *p* = 0.004; (**D**) Intake of CLA-trans10-cis12 and changes in HOMA-IR: *r* = 0.40, *p* = 0.002; (**E**) Intake of C18:2 and changes in HOMA-IR: *r* = −0.36, *p* = 0.006; (**F**) Intake of C18:3 and changes in HOMA-IR: *r* = −0.40, *p* = 0.002; (**G**) Intake of C12:0 and changes in fasting insulin secretion: *r* = −0.31, *p* = 0.03; (**H**) Intake of TRANS 16:1 and changes in fasting insulin secretion: *r* = −0.33, *p* = 0.02.

**Table 1 nutrients-11-00615-t001:** Baseline characteristics of the Study Population. Data are means ± SD, or number (%). *p*-values refer to t-tests for continuous variables and χ*^2^* for categorical variables. The *p*-value calculated for ethnicity distribution is for the comparison between Hispanic versus non-Hispanic categories.

Characteristic	Vegan Group (*n* = 38)	Control Group (*n* = 37)	*p*-Value
Age (years)	52.6 ± 14.7	54.3 ± 9.9	0.56
Sex (number, %)			0.15
Male	2 (5%)	6 (16%)	
Female	36 (95%)	31 (84%)	
Race, (number, %)			
White	15 (39%)	19 (51%)	0.99
Black	20 (53%)	14 (38%)	
Asian, Pacific Islander	0	4 (11%)	
American Indian, Eskimo, Aleut	2 (5%)	0	
N/A: did not disclose	1 (3%)	0	
Ethnicity, (number, %)			
Non-hispanic	33 (87%)	31 (84%)	
Hispanic	3 (8%)	3 (8%)	
N/A: did not disclose	2 (5%)	3 (8%)	
Marital status			
Not married	22 (58%)	19 (51%)	0.66
Married	16 (42%)	17 (46%)	
N/A	0	1 (3%)	
Education			0.49
High school	0	0	
College	17 (45%)	20 (54%)	
Graduate degree	20 (53%)	17 (46%)	
N/A	1 (2%)	0	
Occupation			0.26
Service occupation	5 (13%)	1 (3%)	
Technical, sales, administrative	15 (39%)	10 (27%)	
Professional or managerial	10 (26%)	16 (43%)	
Retired	4 (11%)	6 (16%)	
Other	4 (11%)	4 (11%)	
Medications			
Lipid-lowering therapy (%)	5 (13)	4 (11)	0.99
Antihypertensive therapy (%)	11 (29)	7 (19)	0.31
Thyroid medications (%)	6 (16)	3 (8)	0.48

**Table 2 nutrients-11-00615-t002:** Changes in physical activity, dietary intake, and laboratory and anthropometric variables during the study.

	Control Group		Vegan Group		Treatment Effect	*p*-Value
	Baseline	Week 16	Baseline	Week 16		
**Total physical activity (METs)**	2642 (1476–3809)	2575 (1169–3980)	2207 (1444–2969)	2490 (1586–3395)	+351 (−1143 to +1846)	0.46
**Dietary intake**						
Caloric intake (kcal.day^−1^)	1923 (1627–2219)	1582 (1368–1795) **	1851 (1695–2007)	1450 (1249–1652) ***	−60 (−352 to +233)	0.69
Carbohydrates (% of daily energy)	45.5 (42.6–48.4)	46.6 (42.9–50.4)	46.1 (43.5–48.8)	69.6 (67.3–71.8) ***	+22.3 (+17.7 to +26.9)	<0.001
Fats (% of daily energy)	35.6 (32.3–38.9)	35.0 (31.5–38.4)	36.1 (34.0–38.1)	17.5 (15.5–19.4) ***	−17.9 (−22.3 to −13.6)	<0.001
Proteins (% of daily energy)	16.00 (14.94–17.07)	16.99 (15.45–18.52)	16.77 (15.36–18.19)	12.26 (11.26–13.25) ***	−5.50 (−7.90 to −3.11)	<0.001
Total protein (g/day)	75.4 (65.4–85.4)	66.8 (57.3–76.2)	76.5 (68.8–84.2)	50.8 (42.4–59.3) ***	−17.0 (−30.5 to −3.4)	0.01
Total Carbs (g/day)	215.0 (186.2–243.9)	186.5 (161.2–211.8) *	216.9 (198.2–235.6)	259.9 (222.0–297.8) *	+71.5 (+27.6 to +115.4)	0.002
Total Fat (g/day)	83.9 (65.9–101.9)	66.0 (53.3–78.7) **	77.7 (68.6–86.8)	28.8 (24.5–33.0) ***	−31.0 (−46.2 to −15.7)	<0.001
Cholesterol intake (mg.day^−1^)	290 (220–360)	212 (149–275)	264 (213–315)	6.5 (2.5–10.5) ***	−180 (−278 to −82)	<0.001
Solid Fats (g/day)	33.69 (25.53–41.85)	21.29 (14.47–28.11) **	32.18 (27.49–36.87)	3.77 (2.60–4.94) ***	−16.01 (−25.15 to −6.87)	0.001
**Saturated fatty acids (SFAs)**						
C4:0 butyric acid (g/day)	0.51 (0.33–0.70)	0.28 (0.17–0.39) *	0.51 (0.38–0.65)	0.02 (0.003–0.03) ***	−0.26 (−0.47 to −0.05)	0.02
C6:0 caproic acid (g/day)	0.31 (0.20–0.41)	0.16 (0.10–0.21) **	0.30 (0.22–0.38)	0.01 (0.003–0.02) ***	−0.14 (−0.27 to −0.02)	0.03
C8:0 caprylic acid (g/day)	0.31 (0.20–0.42)	0.16 (0.11–0.21) **	0.33 (0.24–0.43)	0.05 (0.02–0.08) ***	−0.13 (−0.27 to +0.01)	0.07
C10:0 capric acid (g/day)	0.60 (0.39–0.80)	0.30 (0.20–0.39) **	0.59 (0.44–0.74)	0.06 (0.04–0.08) ***	−0.23 (−0.46 to −0.001)	0.05
C12:0 lauric acid (g/day)	1.19 (0.73–1.66)	0.59 (0.35–0.83) *	1.40 (0.86–1.94)	0.29 (0.10–0.48) ***	−0.51 (−1.26 to +0.24)	0.18
C14:0 myristic acid (g/day)	2.14 (1.57–2.71)	1.26 (0.85–1.68) **	2.22 (1.76–2.68)	0.22 (0.13–0.31) ***	−1.13 (−1.83 to −0.43)	0.002
C16:0 palmitic acid (g/day)	13.46 (10.64–16.27)	9.96 (7.75–12.17) **	12.80 (11.30–14.29)	3.60 (2.96–4.24) ***	−5.69 (−8.40 to −2.99)	<0.001
C17:0 margaric acid (g/day)	0.12 (0.09–0.16)	0.07 (0.05–0.10) **	0.09 (0.08–0.11)	0.02 (0.01–0.03) ***	−0.02 (−0.06 to +0.01)	0.20
C18:0 stearic acid (g/day)	6.23 (4.62–7.85)	4.51 (3.10–5.92) **	5.46 (4.77–6.15)	1.20 (0.95–1.45) ***	−2.54 (−3.75 to −1.34)	<0.001
C20:0 arachidic acid (g/day)	0.22 (0.14–0.29)	0.18 (0.13–0.23)	0.18 (0.15–0.21)	0.06 (0.05–0.08) ***	−0.07 (0.14 to −0.01)	0.03
C22:0 behenic acid (g/day)	0.17 (0.09–0.25)	0.16 (0.10–0.22)	0.15 (0.10–0.20)	0.05 (0.03–0.06) ***	−0.09 (−0.20 to +0.02)	0.09
**Monounsaturated fatty acids (MUFAs)**						
C14:1 myristoleic acid (g/day)	0.10 (0.07–0.13)	0.05 (0.03–0.08) **	0.08 (0.06–0.10)	0.002 (0.001–0.002) ***	−0.03 (−0.06 to +0.009)	0.14
C16:1 palmitoleic acid (g/day)	1.05 (0.84–1.26)	0.83 (0.61–1.04) *	1.07 (0.91–1.23)	0.14 (0.10–0.18) ***	−0.71 (−0.97 to −0.45)	<0.001
C18:1 oleic acid (g/day)	28.85 (22.42–35.28)	23.52 (18.67–28.37) *	27.21 (23.51–30.90)	9.13 (7.50–10.75) ***	−12.75 (−18.27 to −7.22)	<0.001
C20:1 gadoleic acid (g/day)	0.28 (0.21–0.35)	0.26 (0.21–0.32)	0.30 (0.25–0.35)	0.10 (0.07–0.13) ***	−0.18 (−0.26 to −0.10)	<0.001
C22:1 erucic acid (g/day)	0.05 (0.02–0.09)	0.09 (0.04–0.14)	0.08 (0.02–0.14)	0.02 (0.01–0.04)	−0.09 (−0.18 to −0.004)	0.04
**Polyunsaturated fatty acids (PUFAs)**						
C18:3 n3 ALA (g/day)	1.98 (1.27–2.69)	1.96 (1.49–2.42)	1.70 (1.37–2.04)	1.15 (0.97–1.33) **	−0.54 (−1.30 to +0.23)	0.16
C18:4 parinaric acid (g/day)	0.01 (0.001–0.02)	0.01 (0.003–0.02)	0.01 (0.003–0.02)	0.0005 (–0.0003–+0.001) *	−0.01 (−0.03 to +0.0016)	0.08
C20:5 EPA (g/day)	0.07 (0.02–0.12)	0.07 (0.03–0.11)	0.06 (0.03–0.10)	0.004 (–0.002–+0.009) **	−0.06 (−0.12 to +0.010)	0.09
C22:5 DPA (g/day)	0.03 (0.02–0.03)	0.03 (0.01–0.04)	0.04 (0.02–0.05)	0.004 (–0.003–0.01) ***	−0.04 (−0.06 to −0.01)	0.004
C22:6 DHA (g/day)	0.12 (0.06–0.17)	0.14 (0.07–0.21)	0.14 (0.07–0.21)	0.02 (–0.01–0.05) **	−0.15 (−0.26 to −0.04)	0.01
Total Omega 3	2.20 (1.47–2.94)	2.21 (1.71–2.71)	1.96 (1.61–2.31)	1.17 (0.99–1.35) ***	−0.79 (−1.59 to −0.004)	0.05
C18:2 linoleic acid (g/day)	18.55 (13.67–23.43)	15.44 (12.60–18.28)	15.89 (13.12–18.66)	9.31 (7.85–10.76) ***	−3.47 (−8.40 to +1.46)	0.16
C18:3 linolenic acid (g/day)	2.03 (1.31–2.75)	2.01 (1.54–2.48)	1.75 (1.41–2.10)	1.16 (0.98–1.34) ***	−0.57 (−1.34 to +0.20)	0.14
C20:4 arachidonic acid (g/day)	0.15 (0.11–0.18)	0.11 (0.08–0.14)	0.14 (0.11–0.17)	0.006 (0.002–0.009) ***	−0.10 (−0.15 to −0.04)	<0.001
**TRANS fats**						
trans 18:1 octadecen (g/day)	1.91 (1.06–2.75)	1.15 (0.66–1.63)	1.67 (1.32–2.02)	0.28 (0.16–0.39) ***	−0.64 (−1.65 to +0.38)	0.21
trans 18:2 octadecad (g/day)	0.31 (0.23–0.39)	0.24 (0.16–0.32)	0.30 (0.26–0.34)	0.07 (0.05–0.09) ***	−0.15 (−0.26 to −0.04)	0.01
trans 16:1 hexadecen (g/day)	0.05 (0.03–0.06)	0.03 (0.02–0.04) **	0.05 (0.04–0.06)	0.004 (0.002–0.005) ***	−0.02 (−0.04 to −0.005)	0.02
CIS9TRANS11	0.10 (0.07–0.13)	0.06 (0.04–0.09)	0.09 (0.08–0.11)	0.003 (0.001–0.08) ***	−0.05 (−0.08 to −0.02)	0.003
CLA TRANS10 CIS12	0.02 (0.02–0.03)	0.01 (0.01–0.02)	0.02 (0.01–0.02)	0.001 (0.008–0.001) ***	−0.009 (−0.016 to −0.003)	0.005
Total Trans Fatty Acids (g/day)	2.29 (1.37–3.21)	1.43 (0.87–2.00)	2.05 (1.67–2.43)	0.35 (0.22–0.48) ***	−0.84 (−1.95 to +0.28)	0.14

Data are means ± SD. Listed *p* values are for interaction between group and time assessed by repeated measures ANOVA. * *p* < 0.05, ** *p* < 0.01 and *** *p* < 0.001 for within-group changes from baseline assessed by paired comparison t tests.

**Table 3 nutrients-11-00615-t003:** Changes in relative dietary intake of Fatty Acids during the study, calculated as percent of total fat.

	Control Group		Vegan Group		Treatment Effect	*p*-Value
	Baseline	Week 16	Baseline	Week 16		
**Saturated fatty acids (SFAs)**						
C4:0 butyric acid (%)	0.561 (0.438–0.684)	0.408 (0.296–0.521)	0.656 (0.515–0.798)	0.046 (0.012–0.080) ***	−0.458 (−0.671 to −0.245)	<0.001
C6:0 caproic acid (%)	0.335 (0.265–0.404)	0.243 (0.172–0.313)	0.388 (0.304–0.472)	0.032 (0.013–0.052) ***	−0.264 (−0.395 to −0.134)	<0.001
C8:0 caprylic acid (%)	0.346 (0.259–0.433)	0.256 (0.187–0.324)	0.426 (0.312–0.539)	0.179 (0.088–0.271) **	−0.156 (−0.334 to +0.022)	0.085
C10:0 capric acid (%)	0.649 (0.525–0.773)	0.461 (0.355–0.568)	0.762 (0.590–0.934)	0.215 (0.136–0.293) ***	−0.360 (−0.603 to −0.117)	0.004
C12:0 lauric acid (%)	1.439 (0.910–1.967)	0.961 (0.586–1.337)	1.762 (1.106–2.417)	1.101 (0.370–1.833)	−0.183 (−1.335 to +0.969)	0.752
C14:0 myristic acid (%)	2.495 (2.096–2.895)	1.963 (1.530–2.396)	2.871 (2.402–3.341)	0.750 (0.460–1.041) ***	−1.589 (−2.364 to −0.813)	<0.001
C16:0 palmitic acid (%)	16.032 (15.145–16.918)	14.930 (13.989–15.870)	16.647 (15.843–17.452)	12.240 (11.690–12.790) ***	−3.305 (−4.816 to −1.795)	<0.001
C17:0 margaric acid (%)	0.146 (0.123–0.170)	0.107 (0.084–0.129) *	0.123 (0.105–0.140)	0.067 (0.020–0.114) *	−0.016 (−0.076 to +0.045)	0.610
C18:0 stearic acid (%)	7.123 (6.371–7.875)	6.449 (5.551–7.347)	7.080 (6.589–7.572)	3.908 (3.577–4.240) ***	−2.498 (−3.410 to −1.586)	<0.001
C20:0 arachidic acid (%)	0.248 (0.205–0.292)	0.265 (0.220–0.310)	0.225 (0.193–0.258)	0.224 (0.194–0.253)	−0.018 (−0.084 to +0.047)	0.580
C22:0 behenic acid (%)	0.209 (0.131–0.286)	0.240 (0.152–0.328)	0.195 (0.134–0.256)	0.163 (0.113–0.212)	−0.064 (−0.190 to +0.062)	0.316
**Monounsaturated fatty acids (MUFA)**						
C14:1 myristoleic acid (%)	0.135 (0.096–0.173)	0.093 (0.055–0.131)	0.103 (0.080–0.126)	0.006 (0.003–0.009) ***	−0.056 (−0.105 to −0.007)	0.026
C16:1 palmitoleic acid (%)	1.296 (1.150–1.442)	1.209 (1.034–1.383)	1.442 (1.247–1.638)	0.482 (0.377–0.587) ***	−0.873 (−0.559 to −1.187)	<0.001
C18:1 oleic acid (%)	33.990 (31.920–36.058)	34.758 (32.622–36.894)	34.766 (33.183–36.349)	31.109 (29.012–33.205) *	−4.427 (−8.037 to −0.816)	0.017
C20:1 gadoleic acid (%)	0.344 (0.292–0.396)	0.417 (0.326–0.509)	0.420 (0.321–0.518)	0.364 (0.246–0.482)	−0.129 (−0.317 to +0.059)	0.175
C22:1 erucic acid (%)	0.071 (0.014–0.127)	0.157 (0.053–0.261)	0.152 (0.026–0.277)	0.095 (0.038–0.153)	−0.143 (−0.332 to +0.046)	0.136
**Polyunsaturated fatty acids (PUFA)**						
C18:2 linoleic acid (%)	22.115 (19.787–24.443)	24.073 (21.827–26.319)	20.048 (18.173–21.924)	32.423 (30.574–34.271) ***	+10.416 (+6.520 to +14.312)	<0.001
C18:3 linolenic acid (%)	2.352 (2.008–2.695)	3.177 (2.559–3.794) *	2.300 (1.897–2.702)	4.338 (3.755–4.921) ***	+1.213 (+0.379 to +2.048)	0.005
C18:3 n3 ALA (%)	2.287 (1.948–2.626)	3.093 (2.478–3.709) *	2.234 (1.838–2.630)	4.280 (3.702–4.858) ***	+1.239 (+0.412 to +2.067)	0.004
C18:4 parinaric acid (%)	0.012 (0.001–0.023)	0.024 (0.006–0.041)	0.024 (0.003–0.044)	0.002 (–0.001–0.006) *	−0.033 (−0.063 to −0.004)	0.027
C20:4 arachidonic acid (%)	0.187 (0.145–0.229)	0.164 (0.124–0.203)	0.189 (0.152–0.226)	0.020 (0.005–0.035) ***	−0.140 (−0.199 to −0.081)	<0.001
C20:5 EPA (%)	0.086 (0.039–0.133)	0.124 (0.053–0.194)	0.108 (0.036–0.181)	0.014 (–0.008–0.037) *	−0.132 (−0.240 to −0.025)	0.017
C22:5 DPA (%)	0.038 (0.021–0.054)	0.054 (0.025–0.084)	0.060 (0.025–0.095)	0.016 (–0.014–0.045)	−0.061 (−0.118 to −0.004)	0.037
C22:6 DHA (%)	0.154 (0.075–0.233)	0.264 (0.125–0.402)	0.234 (0.079–0.389)	0.073 (–0.064–0.210)	−0.271 (−0.522 to −0.019)	0.035
**TRANS fatty acids**						
trans 18:1 octadecen (%)	2.215 (1.272–3.157)	1.626 (1.223–2.028)	2.200 (1.816–2.585)	0.920 (0.561–1.279) ***	−0.692 (−1.802 to +0.419)	0.216
trans 18:2 octadecad (%)	0.376 (0.308–0.444)	0.364 (0.300–0.428)	0.393 (0.352–0.434)	0.233 (0.189–0.277) ***	−0.147 (−0.252 to −0.042)	0.007
trans 16:1 hexadecen (%)	0.059 (0.045–0.072)	0.041 (0.030–0.053)	0.064 (0.049–0.079)	0.012 (0.007–0.016) ***	−0.035 (−0.058 to −0.013)	0.003
Omega–3 (%)	2.576 (2.230–2.921)	3.559 (2.855–4.263) *	2.659 (2.167–3.152)	4.385 (3.791–4.978) ***	+0.742 (−0.216 to +1.701)	0.127
Total CLA (%)	0.139 (0.118–0.160)	0.115 (0.086–0.144)	0.145 (0.122–0.168)	0.013 (0.008–0.019) ***	−0.108 (−0.148 to −0.067)	<0.001
CLA cis9trans12 (%)	0.114 (0.097–0.131)	0.094 (0.070–0.118)	0.121 (0.101–0.140)	0.010 (0.005–0.014) ***	−0.091 (−0.125 to −0.057)	<0.001
CLA trans10cis12 (%)	0.024 (0.020–0.029)	0.020 (0.015–0.025)	0.024 (0.020–0.028)	0.002 (0.001–0.003) ***	−0.017 (−0.024 to −0.011)	<0.001
Total Trans Fatty Acids (%)	2.678 (1.668–3.687)	2.064 (1.607–2.522)	2.694 (2.284–3.105)	1.168 (0.778–1.558) ***	−0.913 (−2.115 to +0.288)	0.133
Solid Fats (%)	38.473 (33.098–43.847)	30.037 (24.940–35.133) *	41.651 (37.762–45.539)	11.921 (9.056–14.816) ***	−21.294 (−29.8 to −12.8)	<0.001

Data are means ± SD. Listed *p* values are for interaction between group and time assessed by repeated measures ANOVA. * *p* < 0.05, ** *p* < 0.01 and *** *p* < 0.001 for within-group changes from baseline assessed by paired comparison t tests.

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
