# Peer review of "Fat Quantity and Quality, as Part of a Low-Fat, Vegan Diet, Are Associated with Changes in Body Composition, Insulin Resistance, and Insulin Secretion. A 16-Week Randomized Controlled Trial"

_nutrients, 2019, doi:10.3390/nu11030615_

Round 1
Reviewer 1 Report
In general, the manuscript is interesting, novel and appropriate for Nutrients.
The manuscript is written and designed, it has a good structure. I have small methodological questions. The methodology is adequate and sufficient. The results are interesting and reliable. The discussion is appropriate. However, I have comments on the manuscript.
Major comments.
1. The title is not appropriate and would be leading to a misunderstanding. The title should include "vegans", because this is a relevant aspect of the study. A "plant-based" diet does not exclude foods of animal origin. Please correct this point.
2. Improve the wording of the paragraph between lines 54 to 56. That statement is not appropriate considering the references used. It is different a study of dietary supplementation, with respect to one of substitution or replacement of components of the diet.
3. The paragraph between lines 57 through 58 must be revised. Question: currently the Western diet is characterized by having a low content of n-6 PUFAs (specifically C18: 2n-6, LA)?
4. Improve the wording of the study objective, and again: the hypothesis should include the word "vegans".
5. The study ended with 75 patients, 67 are women (a study mainly performed in women); I think it is better to withdraw the men from the study, they are only 8. This can generate confusion, and methodologically it is an error. The study is mostly of women.
6. The changes reported by the researchers are very impressive, considering the intervention time (16 weeks), for example; in a vegan group presented; a 106% reduction in fat intake (energy), and a 50% increase in CHO (energy) intake. These are very relevant data for the study, because they allow to have a better understanding of the effects of the intervention on the energy metabolism and of macronutrients in the organism. However, the discussion does not perform a specific analysis on this topic. This aspect must be improved.
7. In the discussion, between lines 210 to 221 describe the results, I think it is not necessary because it was already reported in the writing of results. I suggest to eliminate or reduce.
8. The authors in the discussion use references that are not appropriate, because this is a dietary intervention study, therefore supplementation studies should be left out. If the authors decide to keep them in the discussion, they should indicate what type of studies they are.
Minor comments.
1. In the text they should correct as they are citing the fatty acids, for example: line 57 "palmitic acid", and then on line 237 "palmitic (SFA 16: 0), ....". This fatty acid is cited as "C16: 0". This error occurs with all other fatty acids. Correct this writing error.
2. Line 341 "α-linolenic acids", but the authors on line 293 use the abbreviation ALA. Correct this type of errors throughout the text. The correct and orderly use of abbreviations is relevant is a manuscript.
3. Improve the resolution of the figures, especially figure 2. The size of the letter is very small.
Author Response
We thank reviewer 1 for the insightful comments. We have addressed all of them one-by-one. The changes to the manuscript are highlighted in yellow. Thank you for all your help to improve our manuscript!

Reviewer 2 Report
Comments to the Editor
The manuscript entitled <Fat quantity and quality, as part of a low-fat, plant based, are associated with changes in body composition, insulin resistance, and insulin secretion. A 16-week randomized controlled trial> is well conducted, the results are absolutely interesting and it complies with rigorous scientific criteria, giving the basis to future investigations related with vegan diets and fat intake effects on metabolism.
Comments and suggestions to the authors
· Images quality must be improved, while table editing must be homogenized and well distributed.
· Besides the fatty acid approximation from diet and regime control with patients, did the authors monitored the nutrients’ changes? i.e metals (mainly iron, zinc, manganese, magnesium, or silicon), or vitamins bioavailability or reduction that can affect the lipid metabolism related with lipoprotein composition and transport, and also related with oxidative damage linked to deficient enzymatic activity. And thus, all this associated with insulin resistance changes.
· In this line, how did the authors verified the immunocompetence of the patients during the study? Were there any changes in immune variables? Is it possible that there is some immunological parameter that could be eclipsed, or triggered, by the change in the diet? And thus create some negative, or even more positive, alteration in the patients’ metabolism?
· It is well known that leptin is an important factor regulating the appetite and energy metabolism; also, that disturbances in its signaling are related to changes in adiposity, contributing to excessive body fat. Some recent studies have confirmed the positive effect of vegan diets into the circulating leptin levels. According to this, did the authors evaluate changes in leptin levels? How the energetic balance is affected?
If so, authors should include the information, or at least considered in the discussion section as a present/future explanation related with the metabolic and physical changes observed in patients.
Author Response
We thank reviewer 2 for the insightful comments. We have addressed all of them one-by-one. The changes to the manuscript are highlighted in yellow. Thank you for all your help to improve our manuscript!
